

# The bile acid metabolome in umbilical cord blood and meconium of healthy newborns: distinct characteristics and implications

Chunxia Lu[1,*], Zhiyong Gao[1,*], Siqi Zhang[1], Ke Du[1], Die Xu[1], Wenbin Dong[1], Yujiao Zhang[2] and Xiaoping Lei[1,3]

[1] Division of Neonatology, Department of Pediatrics, The Affiliated Hospital of Southwest Medical University, Luzhou, Sichuan, China
[2] Department of Obstetrics, The Affiliated Hospital of Southwest Medical University, Luzhou, Sichuan, China
[3] Sichuan Clinical Research Center for Birth Defects, The Affiliated Hospital of Southwest Medical University, Luzhou, Sichuan, China
* These authors contributed equally to this work.

Corresponding authors
Yujiao Zhang,
zhangyj0830@163.com
Xiaoping Lei,
leixiaoping2020@swmu.edu.cn

## ABSTRACT

**Objective:** To characterize the bile acid metabolomic profiles of umbilical cord blood and meconium in healthy newborns.

**Methods:** Fifteen healthy newborns, which born in the Obstetrics Department of the Affiliated Hospital of Southwest Medical University between July 1 and August 31, 2023, were selected as study subjects. Umbilical cord blood and meconium samples were collected, and bile acid metabolomics were analyzed using ultra-high performance liquid chromatography-tandem mass spectrometry.

**Results:** The ratio of primary to secondary bile acids in cord blood was significantly higher than in meconium [2.64 (2.49, 5.70) *vs.* 0.99 (0.37, 1.58), Z = −3.80, P < 0.05]. The ratio of unconjugated to conjugated bile acids was notably higher in cord blood than in meconium [0.14 (0.07, 0.18) *vs.* 0.01 (0.01, 0.04), Z = −3.88, P < 0.05]. The ratio of cholic acid to chenodeoxycholic acid in conjugated primary bile acids was significantly lower in cord blood than in meconium [0.59 (0.19, 0.75) *vs.* 2.21 (1.34, 3.04), Z = −4.21, P < 0.05], but the ratio of cholic acid to chenodeoxycholic acid in secondary bile acids was significantly higher in cord blood than in meconium [0.42 (0.21, 0.63) *vs.* 0.03 (0.01, 0.05), Z = −4.54, P < 0.05]. Only three primary bile acids (taurochenodeoxycholic acid, glycochenodeoxycholic acid, and glycochenodeoxycholic acid 3-glucoside in umbilical cord blood) were correlated with their downstream metabolites in meconium (with hyodesoxycholic acid (r = −0.66, P = 0.01), tauro-ω-muricholic acid (r = 0.52, P = 0.048) and ursodeoxycholic acid-7S (r = −0.53, P = 0.04), respectively). In meconium, most of primary bile acids were correlated with their downstream metabolites (P all < 0.05): cholic acid was positively correlated with 3-dehydrocholic acid, taurocholic acid was positively correlated with taurodeoxycholic acid and 3-dehydrocholic acid, glycocholic acid was positively correlated with 3-dehydrocholic acid, chenodeoxycholic acid was positively correlated with glycoursodeoxycholic acid, taurolithocholic acid, and 7-keto lithocholic acid and negatively correlated with isolithocholic acid. Taurochenodeoxycholic acid was positively correlated with taurohyodeoxycholic acid, tauroursodeoxycholic acid, glycoursodeoxycholic acid, taurolithocholic acid, tauro-ω-muricholic acid, and glycohyodeoxycholic acid, while

glycochenodeoxycholic acid was positively correlated with tauroursodeoxycholic acid, glycoursodeoxycholic acid, taurolithocholic acid, and glycohyodeoxycholic acid, and negatively correlated with isolithocholic acid.

**Conclusion:** The bile acid metabolites in umbilical cord blood and meconium differ significantly, and the downstream bile acid metabolites in meconium are predominantly correlated with their upstream bile acids in meconium, but not those bile acids in umbilical cord blood. These findings contribute to a better understanding of bile acid metabolism *in utero* and lay the foundation for future research in this topic.

## INTRODUCTION

Bile acids (BAs), synthesized in hepatocytes using cholesterol, are categorized into primary bile acids (PBAs) and secondary bile acids (SBAs) (*Monte et al., 2009*; *Russell, 2003*; *Zheng et al., 2021*). The originally generated BAs are the unconjugated PBAs, namely cholic acid (CA) and chenodeoxycholic acid (CDCA). These unconjugated BAs have the capacity to conjugate with glycine and taurine, resulting in conjugated BAs. Subsequently, under the influence of intestinal flora, these conjugated BAs generate SBAs. As the principal constituent of bile, BAs play a crucial role in enhancing the emulsification of fats, thereby aiding in the digestion and absorption of fats and fat-soluble vitamins (*Begley et al., 2005*; *Monteiro-Cardoso, Corlianò & Singaraja, 2021*). They also can be as metabolic signaling molecules, playing a crucial role in regulating lipid and glucose metabolism (*Hansen, Sonne & Knop, 2014*; *Lefebvre et al., 2009*).

Regarding the bile acid profiles during fetal development, CDCA conjugates become prevalent in the first months of life (*Brites et al., 1998*; *Niijima, 1985*; *Polkowska et al., 2001*; *Suchy et al., 1981*; *Zöhrer et al., 2016*), in contrast to that in healthy older children and adults (*Brites et al., 1998*; *Jahnel et al., 2015*; *Serrano et al., 1998*). In addition, in the first days of life, the conjugation of BAs is predominantly done with taurine (*Challacombe, Edkins & Brown, 1975*; *Zöhrer et al., 2016*) and soon afterward with glycine (*Niijima, 1985*). Thus, glycocholic acid (GCA) is the main bile acid in the neonatal period, and between 1 and 3 months old, this predominance becomes from glycochenodeoxycholic acid (GCDCA) (*Niijima, 1985*). Moreover, the elevation of primary to secondary BAs is also characteristic of infants with less than 1 year of age (*Niijima, 1985*). *Niijima (1985)* also described that, though GCA predominates in the neonatal period, after 1–3 months GCDCA starts to predominate.

Increasing studies had highlighted the association of BAs with various aspects of fetal development, including physical growth, respiratory and the nervous system. Notably, PBA levels in umbilical cord blood (UCB) exhibit a significant positive correlation with the risk of infants being small for gestational age or having a low birth weight (*Xie et al., 2024*), suggesting a potential role in regulating fetal growth and development. Animal studies have further elucidated the involvement of BAs in regulating respiratory rhythm and even

directly influencing respiratory function (*Wang et al., 2013*; *Zhao et al., 2014*). Additionally, serum bile acids have been found to inhibit the production of pulmonary surfactant, which can lead to the onset of respiratory distress syndrome (*Yu et al., 2014*). In the context of fetal rat models, serum total bile acid levels (TBAs) have been observed to correlate positively with neuron-specific enolase concentration (*Tan & Ding, 2007*). These levels have also been associated with adverse outcomes such as increased blood-brain barrier permeability (*Quinn et al., 2014*), neuroinflammation (*McMillin et al., 2017*), neurological decline (*McMillin et al., 2016*), and cerebellar sensorimotor deficits (*Henriksen et al., 2022*). Furthermore, elevated SBA levels and the SBA/PBA ratio have been significantly linked to poorer cognitive abilities (*MahmoudianDehkordi et al., 2019*).

Although previous investigations have observed many biological impacts of TBAs on the fetus, the complexity of the bile acid metabolome, comprising numerous active metabolites, poses a challenge to identify a specific active component in the comprehensive network of bile acid pool. Establishing the bile acid metabolomic profile of healthy newborns serves as a foundation for future research in this field. The fetal period, as a unique phase in life, is dynamically influenced by liver metabolism (*Grijalva & Vakili, 2013*) and the mother's bile acids (*Hassan & Subbiah, 1980*) in terms of bile acid metabolomics. Furthermore, there is a correlation between the newborn's metabolome and that of the mother since pregnancy, suggesting that the newborn is plausibly "programmed" by the maternal metabolism (*Perrone et al., 2020*). Meconium, composed primarily of shed intestinal epithelial cells and swallowed amniotic fluid during fetal life, contains high concentrations of BAs metabolized and excreted by the liver (*Kumagai et al., 2007*). Given that PBAs are predominantly metabolized and synthesized in the liver and subsequently converted into SBAs by intestinal flora upon entering the intestine, the immature intestinal flora during the fetal period suggests that bile acid metabolomics in meconium may exhibit greater stability than blood circulation. This stability offers significant value in guiding subsequent research.

To date, limited studies have been conducted on the BAs present in UCB and fecal samples from healthy newborns. The aim of this study was to comprehensively describe the bile acid metabolomic characteristics of UCB and meconium, as well as to explore the correlation between the upstream and downstream metabolites of BAs in both UCB and meconium. Through this, we sought to obtain a more thorough and complete understanding of the bile acid metabolomic profile of healthy newborns. Furthermore, this establishes a foundation for further investigation into the relationship between BAs and the development of fetal physical growth, respiratory function, and the nervous system.

## MATERIALS AND METHODS

### Study subjects

Fifteen healthy newborns, born in the Obstetrics Department of the Affiliated Hospital of Southwest Medical University between July 1 and August 31, 2023, were selected for this study. A total of 19 subjects were recruited for this study. Following recruitment, four neonates were excluded from the study because only cord blood or meconium was collected, resulting in incomplete specimens. The study protocol was approved by the

Affiliated Hospital of Southwest Medical University's Ethics Committee (Ethical Application Ref: KY2023182). There was no further changes or amendments made after protocol approval. In addition, we received written informed consent from participants of our study prior to the collection of biological samples. All samples collected are numbered so as not to reveal the subject's name and other privacy details.

## Inclusion and exclusion criteria

### Inclusion criteria

1) Mothers without any diseases or complications during pregnancy and aged 18–45 years. 2) Gestational age at birth exceeding 37 weeks. 3) Natural conception. 4) Singleton birth. 5) Cesarean section delivery. 6) Willingness to participate in the study.

### Exclusion criteria

1) Newborns with existed medical conditions at birth or newly occurred medical conditions in the first day of life. 2) No intention to participate in the study.

## Clinical data collection

Detailed records were kept of the mothers' age, height, weight, medication history during pregnancy, gestational age, and other relevant information before delivery. After delivery, the newborns' gender, Apgar score, height, weight, and head circumference were measured and documented.

## Sample collection

UCB: UCB was collected from the newborns within five minutes after delivery and before the placenta was delivered. The samples were then preprocessed uniformly and stored at −80 °C until further testing. Meconium: Researchers, wearing disposable sterile gloves, collected fresh meconium (the first excretion within 24 h of birth) into specialized sterile stool boxes. The meconium samples were quickly frozen in liquid nitrogen and stored at −80 °C until testing.

## Sample preprocessing

UCB: Initially, 50 μL of serum sample was added to a 1 mL 96-well plate, followed by the addition of 400 μL of an acetonitrile/methanol (v/v = 8:2) mixed solvent containing an internal standard. The 96-well plate was then centrifuged at 10 °C and 650 rpm for 20 min. Subsequently, 250 μL of the supernatant was transferred to a 350 μL microplate and lyophilized. The dried sample was reconstituted with 40 μL of acetonitrile/methanol (80/20) solution, centrifuged again under the same conditions, and then 60 μL of deionized water was added followed by another centrifugation step. The microplate was then frozen at −20 °C for 20 min, centrifuged at 4 °C and 4,000 g for 30 min, and finally, a 5 μL injection volume was prepared for testing.

Meconium: Initially, approximately 10 mg of meconium sample was carefully weighed and placed into an Eppendorf centrifuge tube. Subsequently, about 25 mg of pre-cooled grinding beads were added to facilitate homogenization. Following this, 200 μL of a mixed solvent consisting of acetonitrile and methanol in an 8:2 ratio, containing 10 μL of an

internal standard, was incorporated. After thorough homogenization, the mixture was centrifuged at 4 °C and 13,500 rpm for 20 min. From the resulting supernatant, 20 μL was extracted and diluted with 80 μL of a 1:1 mixture of acetonitrile/methanol (80/20) and ultrapure water for further analytical testing. The final injection volume for analysis was set at 5 μL.

## Quantification of bile acid metabolites and quality assurance

For the determination of bile acid metabolites, UPLC-MS/MS was employed. The instrument parameters were fine-tuned for accurate measurement of these metabolites in UCB and meconium samples. To ensure data quality, stable isotope-labeled internal standards and mixed quality control samples were utilized. Additionally, blank solvent samples were run to guarantee optimal instrument performance.

## Statistical analysis methodology

Data analysis was conducted using SPSS 27.0.1. The Shapiro-Wilk test was applied to assess data normality. For normally distributed measurement data, the independent sample t-test was used, and results are expressed as mean ± standard deviation (Mean ± SD). For data that did not conform to a normal distribution, the Mann-Whitney U test was employed, with results presented as median (first Quartile- third Quartile). Correlation analysis was conducted using the Pearson linear method for normally distributed variables and the Spearman rank method for non-normally distributed variables or hierarchical data. In the present study, we only reported the correlations between bile acids that exhibit upstream-downstream relationships or those potentially involved in the physiological processes of the enterohepatic circulation. Statistical significance was determined at a $P$-value less than 0.05.

# RESULTS

## Baseline information of study subjects

As shown in Table 1, the average age of the subjects' mothers was 29.5 ± 3.7 years old, and the average gestational age of the newborns was 39.3 ± 0.8 weeks.

## Characteristics of bile acid metabolites in UCB and meconium of healthy newborns

### Composition of bile acid metabolites in UCB and meconium

In UCB of healthy newborns, a total of 30 bile acid metabolites were identified. These metabolites were categorized into 8 (26.67%) types of PBAs and 22 (73.33%) types of SBAs. In terms of content, PBAs were found to be the predominant component with a concentration of 2,074.20 (1,508.03, 2,969.55) nmol/L, whereas SBAs had a concentration of 772.18 (580.64, 892.19) nmol/L (Z = −4.63, $P$ < 0.05).

Regarding the components, unconjugated BAs and conjugated BAs in UCB were present at concentrations of 325.81 (222.87, 438.76) nmol/L and 2,375.99 (1,956.97, 3,740.23) nmol/L (Z = −4.67, $P$ < 0.05), respectively. Within the conjugated BAs, taurine-conjugated BAs were the main component with a concentration of 1,583.11

**Table 1 Baseline information of study subjects.**

|  | n = 15 |
|---|---|
| **Mothers** | |
| Age (years) | 29.5 ± 3.7 |
| Body mass index (kg/m$^2$) before delivery | 26.8 ± 2.3 |
| Admission glucose (mmol/L) | 4.4 ± 0.6 |
| Prenatal antibiotics (Yes/No) | 0/15 |
| Smoking (Ever/Never) | 0/15 |
| **Newborns** | |
| Gender (Male/Female) | 9/6 |
| First-minute Apgar score | 10 (10, 10) |
| Weight (g) | 3,347 ± 348 |
| Gestation week (Weeks) | 39.3 ± 0.8 |
| Length (cm) | 50.5 ± 1.1 |
| Head circumference (cm) | 33.5 ± 1.2 |

(1,235.97, 2,389.81) nmol/L, followed by glycine-conjugated BAs at 820.63 (587.42, 1,350.42) nmol/L ($Z = -2.72$, $P < 0.05$). Individual metabolites with the highest contents included taurochenodeoxycholic acid (TCDCA) (697.90 nmol/L, 27.2%), taurocholic acid (TCA) (491.04 nmol/L, 19.1%), and GCDCA (488.65 nmol/L, 19.0%).

In meconium, a total of 35 bile acid metabolites were identified. These were classified into six (17.14%) types of PBAs and 29 (82.86%) types of SBAs. The meconium was composed of SBAs with a concentration of 3,093.78 (2,223.83, 4,048.72) nmol/g, PBAs were present at a concentration of 2,825.39 (1,472.38, 5,065.02) nmol/g ($P > 0.05$). The PBAs/SBAs ratio in UCB was significantly higher than that in meconium [2.64 (2.49, 5.70) *vs* 0.99 (0.37, 1.58), $Z = -3.80$, $P < 0.05$] (Fig. 1A).

In terms of composition, the unconjugated bile acid contents in meconium were only 118.12 (44.64, 171.40) nmol/g, whereas conjugated BAs were the main component at 5,292.88 (4,312.29, 7,719.61) nmol/g ($Z = -4.67$, $P < 0.05$). The ratio of unconjugated BAs to conjugated BAs in UCB was significantly higher than that in meconium [0.14 (0.07, 0.18) *vs.* 0.01 (0.01, 0.04), $Z = -3.88$, $P < 0.05$] (Fig. 1B). Among the conjugated BAs, taurine-conjugated BAs were the dominant type with a concentration of 4,527.50 (3,692.15, 6,622.30) nmol/g, followed by glycine-conjugated BAs at 731.62 (584.60, 986.82) nmol/g ($Z = -4.13$, $P < 0.05$). The individual metabolite contents were highest for TCA (1,628.03 nmol/g, 28.3%), taurohyocholic acid (THCA) (1,522.10 nmol/g, 26.4%), and TCDCA (986.06 nmol/g, 17.1%).

### The composition of CA metabolites and CDCA metabolites in UCB and meconium

Bile acids were categorized into CA metabolites and CDCA metabolites based on their origins. In UCB, we observed that PBAs were predominantly derived from CDCA. Specifically, the PBAs originating from CA were 676.73 (452.89, 947.40) nmol/L, whereas

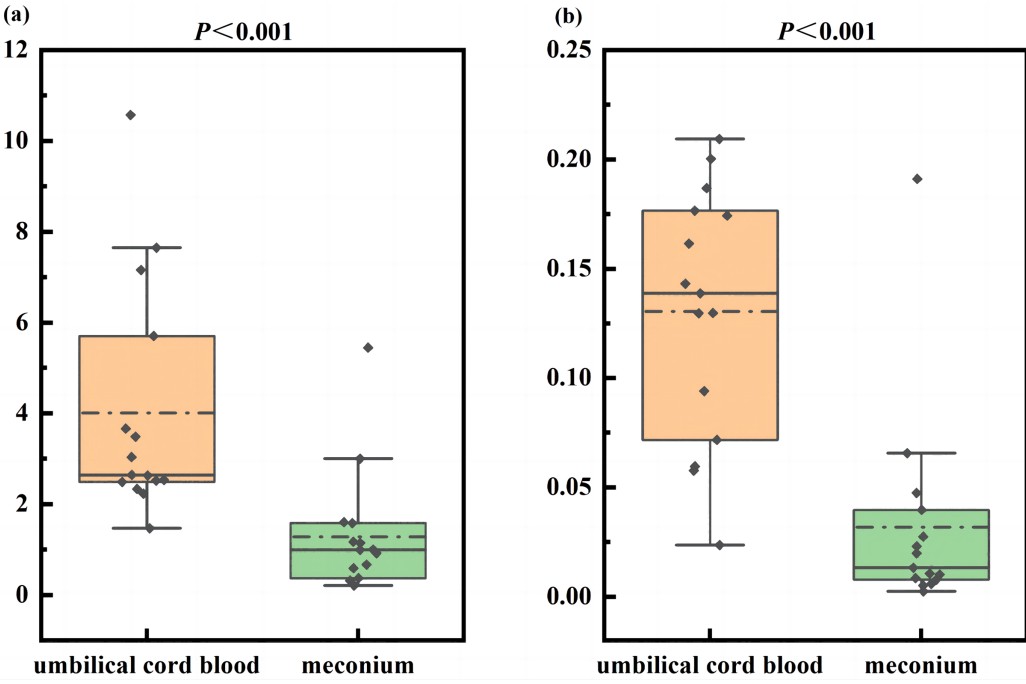

**Figure 1** **The composition ratio of primary bile acids to secondary bile acids and unconjugated bile acids to conjugated bile acids in umbilical cord blood and meconium.** (A) The composition ratio of primary bile acids to secondary bile acids in umbilical cord blood and meconium; (B) the composition ratio of unconjugated bile acids to conjugated bile acids in umbilical cord blood and meconium.

that from CDCA were significantly higher at 1,187.31 (1,032.32, 2,319.80) nmol/L ($Z = -2.97$, $P < 0.05$). Similarly, SBAs in UCB were mainly derived from CDCA, with CA-derived SBAs at 178.45 (112.37, 282.47) nmol/L compared to 507.32 (349.83, 634.25) nmol/L for CDCA-derived SBAs ($Z = -4.13$, $P < 0.05$).

In meconium, our analysis revealed no statistically significant between the PBAs content derived from CDCA and those from CA [1,065.83 (486.68, 1,560.61) nmol/g *vs.* 2,022.57 (652.54, 3,257.16) nmol/g, $P > 0.05$]. But there was a significantly lower ratio of CA to CDCA-derived PBAs in UCB than in meconium [0.59 (0.19, 0.75) *vs.* 2.21 (1.34, 3.04), $Z = -4.21$, $P < 0.05$] (Fig. 2A).

Regarding SBAs in meconium, the contents derived from CA were substantially lower than from CDCA [89.22 (24.02, 170.37) nmol/g *vs.* 2,948.27 (2,133.02, 3,864.00) nmol/g, $Z = -4.67$, $P < 0.05$]. Although both were predominantly derived from CDCA, the ratio of CA to CDCA-derived SBAs in UCB was significantly higher than that in meconium [0.42 (0.21, 0.63) *vs.* 0.03 (0.01, 0.05), $Z = -4.54$, $P < 0.05$] (Fig. 2B).

## Correlation of bile acid metabolites between UCB and meconium in healthy newborns

### Correlation between PBAs in UCB and meconium

Among healthy newborns, our analysis revealed no statistically significant correlation between the levels of PBAs in UCB and those found in meconium ($P > 0.05$).

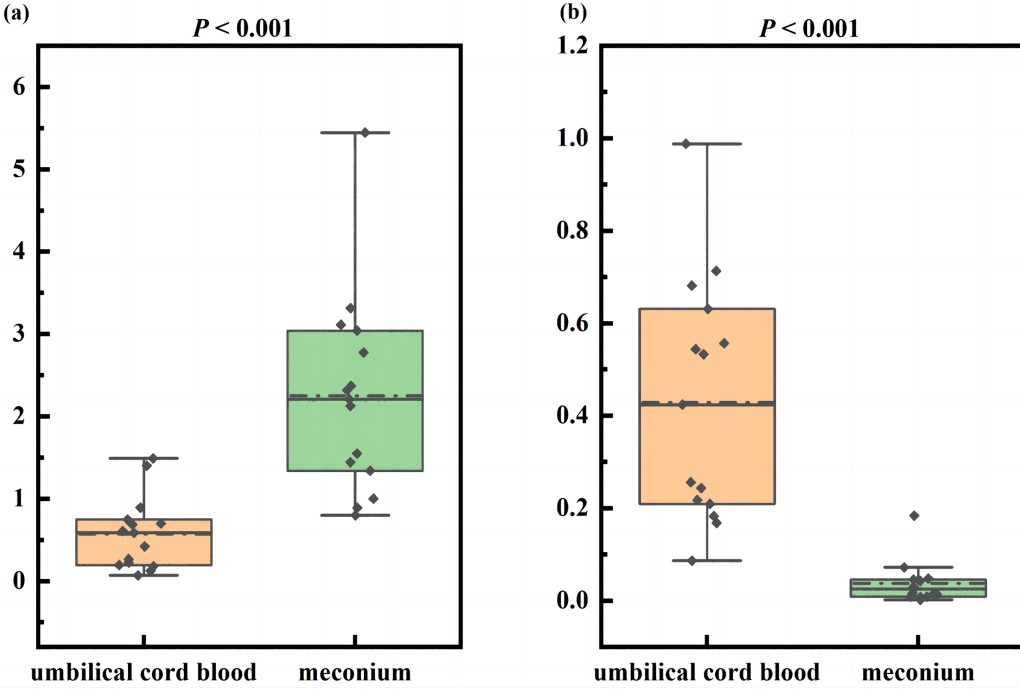

**Figure 2 The ratio of cholic acid-derived metabolites to chenodeoxycholic acid-derived metabolites in umbilical cord blood and meconium.** (A) The ratio of cholic acid-derived primary bile acids to chenodeoxycholic acid-derived primary bile acids in umbilical cord blood and meconium; (B) the ratio of cholic acid-derived secondary bile acids to chenodeoxycholic acid-derived secondary bile acids in umbilical cord blood and meconium.

### Correlation between PBAs in UCB and its responding downstream SBAs in meconium

As illustrated in Fig. 3, most PBAs in UCB did not correlate with their downstream metabolites detected in meconium. Only a few PBAs in UCB were correlated with their downstream metabolites in meconium: A negative correlation between TCDCA in UCB and its responding secondary bile acid, hyodeoxycholic acid (HDCA) in meconium ($r = -0.66$, $P = 0.01$). Glycochenodeoxycholic acid-3-glucuronide (GCDCA-3-Glu) in UCB exhibited a negative correlation with ursodeoxycholic acid-7-sulfate (UDCA-7S) in meconium ($r = -0.53$, $P = 0.04$), while a positive correlation was observed between GCDCA in UCB and tauro-ω-muricholic acid (TωMCA) in meconium ($r = 0.52$, $P = 0.048$).

### Correlation between PBAs and their downstream SBAs in meconium

As depicted in Fig. 4, there exists a noteworthy correlation between most PBAs and their responding SBAs in meconium:

1) CA exhibits a positive correlation with its 3-dehydrocholic acid (3-DHCA) ($r = 0.71$, $P = 0.003$).

2) TCA demonstrates a positive association with taurodeoxycholic acid (TDCA) ($r = 0.66$, $P = 0.01$) and 3-DHCA ($r = 0.62$, $P = 0.01$).

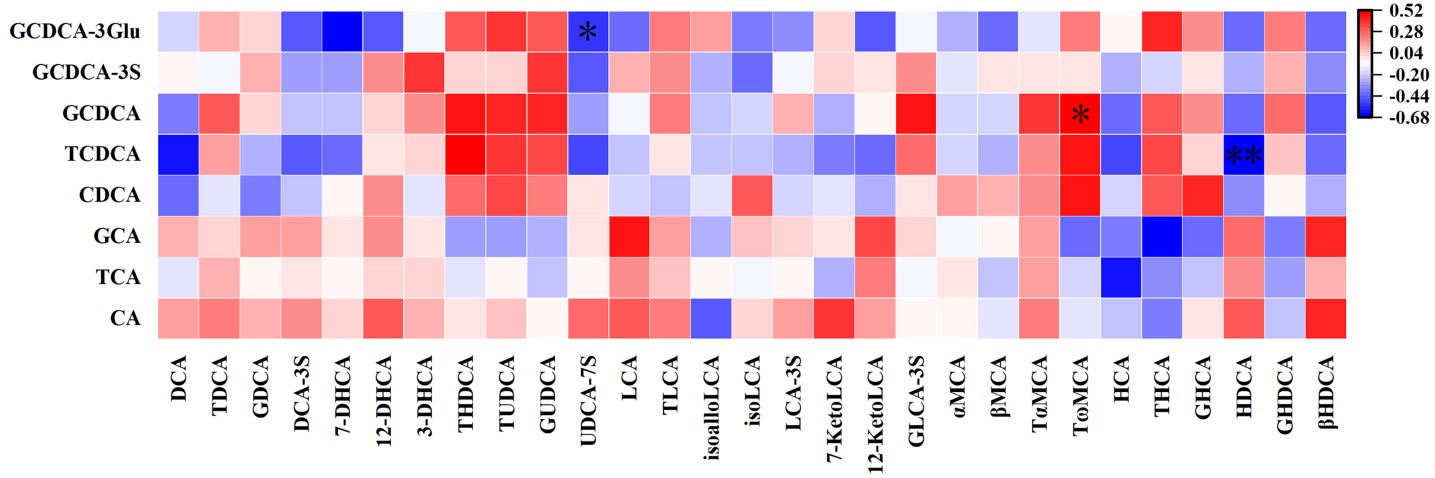

**Figure 3** Correlation between primary bile acids in umbilical cord blood and their responding downstream secondary bile acids in meconium. *P < 0.05; **P < 0.01.

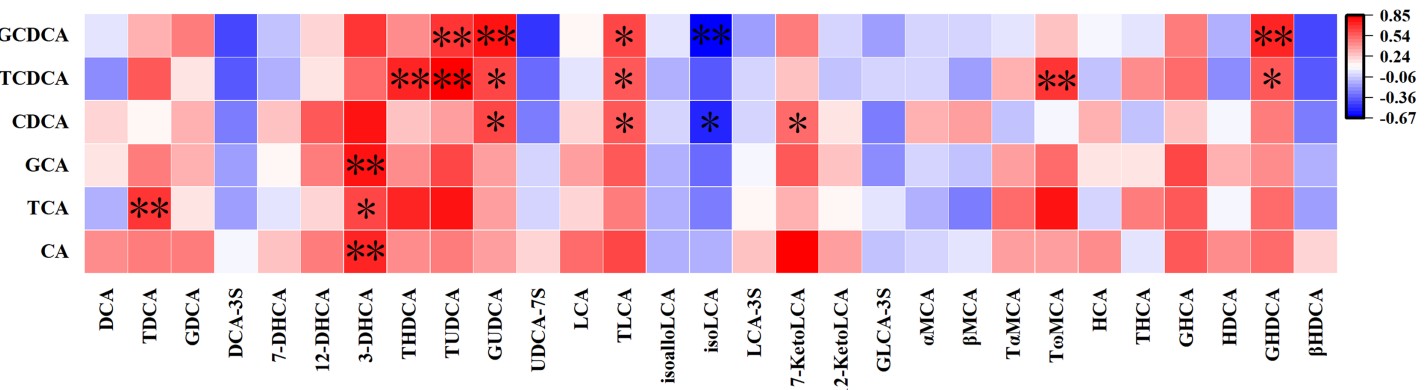

**Figure 4** Correlation between primary bile acids and their downstream secondary bile acids in meconium. *P < 0.05; **P < 0.01.

3) GCA is positively correlated with its downstream 3-DHCA (r = 0.76, P = 0.001).

4) CDCA shows a positive correlation with glycoursodeoxycholic acid (GUDCA) (r = 0.60, P = 0.02), taurolithocholic acid (TLCA) (r = 0.56, P = 0.03), and 7-ketolithocholic acid (7-KetoLCA) (r = 0.52, P = 0.045), while being negatively correlated with isolithocholic acid (isoLCA) (r = −0.53, P = 0.04).

5) TCDCA is positively correlated with taurohyodeoxycholic acid (THDCA) (r = 0.74, P = 0.002), tauroursodeoxycholic acid (TUDCA) (r = 0.84, P = 0.0001), GUDCA (r = 0.61, P = 0.02), TLCA (r = 0.58, P = 0.02), TωMCA (r = 0.65, P = 0.01), and glycohyodeoxycholic acid (GHDCA) (r = 0.55, P = 0.04).

6) GCDCA demonstrates a positive correlation with TUDCA (r = 0.65, P = 0.01), GUDCA (r = 0.78, P = 0.001), TLCA (r = 0.61, P = 0.02), and GHDCA (r = 0.70, P = 0.003), while being negatively correlated with isoLCA (r = −0.66, P = 0.01).

These findings highlight the complex metabolic relationships between primary and secondary BAs in meconium.

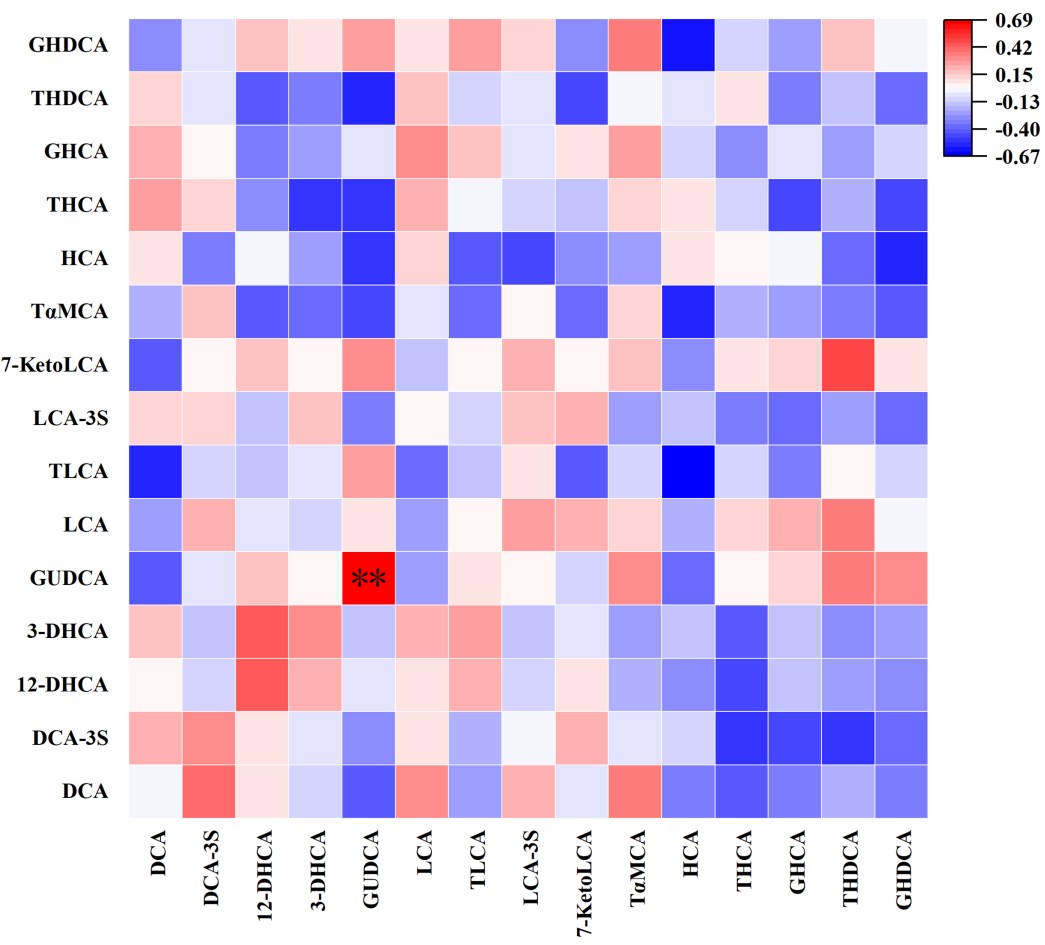

**Figure 5 Correlation between secondary bile acids shared by meconium and umbilical cord blood.**
**$P < 0.01$.

### Correlation between SBAs shared by meconium and UCB

As illustrated in Fig. 5, the majority of SBAs that are common to both meconium and UCB did not demonstrate a statistically significant correlation. Only GUDCA exhibited a positive correlation between UCB and meconium ($r = 0.69$, $P = 0.004$).

## DISCUSSION

The findings of the present study reveal distinct distribution traits for bile acid metabolites and exhibit less relationship between UCB and meconium. However, a correlation was observed between PBAs and SBAs in meconium, suggesting that the fetal intestine is capable of synthesizing SBAs.

Our research indicates that the predominant bile acid metabolites found in UCB are PBAs, aligning with a previous study (*Shoda et al., 1988*). Within the maternal-infant bile acid exchange system, PBAs are primarily transferred from the fetus to the mother (*Marin et al., 2008*), leading us to hypothesize that the PBAs detected in UCB are likely derived from fetal biosynthetic processes. A significant positive correlation has been demonstrated

between PBA levels in UCB and the risk of having a small-for-gestational-age infant or low birth weight (*Xie et al., 2024*). This suggests that PBAs may play a role in regulating fetal growth and development. By elucidating the distribution patterns of bile acids in UCB of healthy newborns, our study provides a foundation for further exploration into the role of BAs in fetal growth and developmental regulation.

In this study, we detected large amounts of SBAs in meconium, aligning with a previous study (*Kumagai et al., 2007*). However, no statistically significant distinction was observed between the concentrations of PBAs and SBAs in meconium. This may be a result of the influence of the fetal gut flora. Although some studies have confirmed the existence of fetal intestinal flora (*Satokari et al., 2009*), it has been traditionally believed that the fetal intestine lacks the flora necessary for bile acid metabolism and has limited capacity to produce SBAs. However, our findings suggest that the fetal intestine possesses the ability to synthesize SBAs, which are likely generated from PBAs under the influence of intestinal flora. This raises the possibility that the fetal intestine harbors the flora required for bile acid metabolism. Given that intestinal flora play a role in regulating various pathophysiological processes, including growth and development, as well as the onset and progression of metabolic diseases, and that BAs directly impact glucose and lipid metabolism, our results offer valuable insights into the regulatory mechanism of bile acid-intestinal flora interactions in the intrauterine origin of metabolic diseases.

Moreover, our findings indicate that the bile acid metabolites in UCB and meconium are predominantly taurine-conjugated BAs. This is consistent with previous research showing that taurine conjugates predominate in fetal liver bile acids (*Itoh & Onishi, 2000*). Since unconjugated BAs can be hepatotoxic, taurine conjugates, which are more polar than their glycine counterparts (*Gu et al., 1992*), may serve as a protective mechanism for the fetus. Taurine, the most abundant free amino acid in the human body, plays a crucial role in processes like bile acid conjugation, calcium homeostasis, osmotic regulation, and membrane stability (*Marcinkiewicz & Kontny, 2014*). It effectively reduces plasma and liver cholesterol concentrations, participating in the cholesterol 7α-hydroxylase (CYP7A1) -mediated regulatory mechanism of cholesterol and bile acid homeostasis (*Chen, Guo & Chang, 2012*). Therefore, we hypothesize that taurine-conjugated BAs in UCB and meconium may contribute to regulating cholesterol and bile acid homeostasis.

In healthy full-term infants, the SBAs present in UCB and meconium are predominantly derived from CDCA, a finding that differs from the source of bile acid metabolism after birth (*Grijalva & Vakili, 2013*). According to a previous study, fetal bile acid synthesis predominantly follows the alternative pathway of CDCA synthesis (*Setchell et al., 1988*). When this alternative pathway dominates, it leads to the production of more hydrophilic bile acids, subsequently reducing intestinal cholesterol and fat absorption (*Wang et al., 2003*). Previous studies have indicated that SBAs in fetal blood could originate from maternal bile transported *via* the placenta (*Colombo et al., 1985*; *Itoh et al., 1982*; *Sigurdsson et al., 2016*). However, the composition of SBAs in UCB and meconium was consistent in this study, suggesting that the source of fetal blood SBAs may involve

intestinal reabsorption. Furthermore, our study observed a weak correlation between PBAs in UCB and their counterparts and downstream SBAs in meconium. In contrast, a significant correlation exists between PBAs and their downstream SBAs in meconium. These findings indirectly support the notion that the newborn's intestine has the capability to synthesize SBAs, aligning with the research conducted by *Shoda et al. (1988)*.

In addition, the present study demonstrated that GUDCA exhibited a positive correlation between UCB and meconium. It has also been reported that enhanced fecal BA loss is accompanied by enhanced hepatic BA synthesis (*Herrema et al., 2010*; *Huang et al., 2019*; *Out et al., 2011*). This may be the mechanism by which the fetus regulates the balance of its own BAs in order to maintain the dynamic balance of the various BAs in the fetus. Furthermore, it has been demonstrated that GUDCA can diminish blood glucose, cholesterol, and triglyceride concentrations in the livers and serum of mice (*Chen et al., 2023*). It is hypothesized that the fetus may be involved in regulating its glycolipid metabolism by modulating the balance of regulated GUDCA. However, this hypothesis requires further investigation through additional animal experiments.

Compared to previous studies (*Kumagai et al., 2007*; *Xie et al., 2024*), this may be the first report associating bile acid profiles in UCB with those in meconium in healthy infants. The strength of this study is to use samples from the same individuals, nullifying the interspecies differences in the bile acid physiology affecting other studies' results. However, several limitations of the present study should be noted. This study did not include bile acid metabolomic analysis of pregnant women's peripheral blood and cannot analyze the correlation of serum BAs between maternal and fetal circulation. However, it has been reported that the metabolic profile of the newborn correlates with that of the mother at 3 weeks prior to delivery (*Perrone et al., 2020*). Moreover, a previous report has indicated that BAs in UCB can be influenced by bile acids in the maternal circulation (*Itoh et al., 1982*). It is also important to note that our study was conducted at a single center with a limited sample size (15 newborns). To confirm the reliability of our findings, further large-scale, multi-center studies are warranted. In the future research, we can also explore the implications of bile acid metabolism in different populations of newborns or investigate the long-term health outcomes related to the findings. Additionally, as we did not test intestinal flora, we cannot provide direct evidence of their involvement in bile acid metabolism.

## CONCLUSIONS

In conclusion, the bile acid metabolites in UCB and meconium differ significantly in healthy newborns, and the downstream bile acid metabolites in meconium are predominantly correlated with their upstream BAs in meconium, but not those BAs in UCB, this indirectly suggests the presence of SBA-synthesizing capability in the intestine.

## ACKNOWLEDGEMENTS

Thank you to all the medical staff of the Neonatology Department of the Affiliated Hospital of Southwest Medical University.

### Funding

This work was supported by the Science and Technology Strategic Cooperation Programs of Luzhou Municipal People's Government and Southwest Medical University, Grant No. 2021LZXNYD-J21 to Xiaoping Lei. The funders had no role in study design, data collection and analysis, decision to publish, or preparation of the manuscript.

### Grant Disclosures

The following grant information was disclosed by the authors:
Science and Technology Strategic Cooperation Programs of Luzhou Municipal People's Government and Southwest Medical University: 2021LZXNYD-J21.

### Competing Interests

The authors declare that they have no competing interests.

### Author Contributions

- Chunxia Lu conceived and designed the experiments, performed the experiments, analyzed the data, prepared figures and/or tables, specimen collection, and approved the final draft.
- Zhiyong Gao conceived and designed the experiments, performed the experiments, analyzed the data, prepared figures and/or tables, specimen collection, and approved the final draft.
- Siqi Zhang analyzed the data, prepared figures and/or tables, and approved the final draft.
- Ke Du performed the experiments, authored or reviewed drafts of the article, specimen collection, and approved the final draft.
- Die Xu performed the experiments, analyzed the data, prepared figures and/or tables, authored or reviewed drafts of the article, and approved the final draft.
- Wenbin Dong conceived and designed the experiments, authored or reviewed drafts of the article, and approved the final draft.
- Yujiao Zhang conceived and designed the experiments, performed the experiments, authored or reviewed drafts of the article, and approved the final draft.
- Xiaoping Lei conceived and designed the experiments, prepared figures and/or tables, and approved the final draft.

### Human Ethics

The following information was supplied relating to ethical approvals (*i.e.*, approving body and any reference numbers):

Clinical Trial Ethics Committee of the Affiliated Hospital of Southwest Medical University granted Ethical approval to carry out the study within its facilities (Ethical Application Ref: KY2023182).

## Data Availability

The data is available at MetaboLights: MTBLS10266, MTBLS10267 and in the Supplemental Files.

## Supplemental Information

Supplemental information for this article can be found online at http://dx.doi.org/10.7717/peerj.18506#supplemental-information.

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
