# Peer review of "The bile acid metabolome in umbilical cord blood and meconium of healthy newborns: distinct characteristics and implications"

_PeerJ, doi:10.7717/peerj.18506_

## Round 0.1 · original submission · Major Revisions

All 3 reviewers have provided good comments - please address them all. We look forward to your revision.

·

Basic reporting

Clear and unambiguous, professional English used throughout: The article could benefit from simplifying some complex sentences for better readability. For example, "This phenomenon leads us to hypothesize that the PBAs detected in UCB likely originate from the fetus's own biosynthetic processes".

Literature references, sufficient field background/context provided: The discussion could better link findings to existing studies.

Self-contained with relevant results to hypotheses
Relevant results to hypotheses: The implications of the correlations between metabolites could be explained more clearly.

Experimental design

The research question could be enhanced by explicitly stating the potential impact or broader implications of the findings in the introduction. This would make the relevance and significance of the study more apparent.

It is stated how research fills an identified knowledge gap: The manuscript could more clearly articulate how it advances the current understanding compared to previous studies. A brief summary of how the findings specifically contribute to the field would strengthen this aspect.

Rigorous investigation performed to a high technical & ethical standard: Providing more detail on the criteria for participant selection and any potential biases or limitations in the study design would further reinforce the study’s rigor.

Methods described with sufficient detail & information to replicate: Some steps, such as the reason behind specific statistical tests or the selection of particular metabolites for analysis, could be explained to ensure replicability.

Validity of the findings

Impact and novelty not assessed. Meaningful replication encouraged where rationale & benefit to literature is clearly stated: The manuscript could benefit from more emphasis on the potential for replication and how this could contribute to the broader literature.
All underlying data have been provided; they are robust, statistically sound, & controlled: consider describing how any potential confounding factors were controlled during data collection and analysis.
Conclusions are well stated, linked to original research question & limited to supporting results: It could be enhanced by acknowledging any limitations and suggesting areas for future research.

Additional comments

Sample Size: The relatively small sample size (15 newborns) could limit the generalizability of the findings.

Ethical Considerations: It would be useful to include a brief statement on how informed consent was obtained and how the study ensured the confidentiality and well-being of the participants.

Future Research Directions: Exploring the implications of bile acid metabolism in different populations of newborns or investigating the long-term health outcomes related to the findings could be added.

·

Basic reporting

The article is well written and use clear, unambiguous, technically correct text.

In the background authors should include more references regarding primary and secondary bile acid in physiological newborns and also taking into account the influence of the maternal metabolome on the newborn.

The structure of the article is conform to ‘standard sections’.

This manscript is ‘self-contained,’ represent an appropriate ‘unit of publication’, and include all results relevant to the hypothesis.

Experimental design

Dear Authors,
I read your manuscript with great interest. I would suggest some corrections to improve your work:

In the line 110, authors do not declare delivery mode of the newborns. This must be reported as the delivery mode can influence the intestinal microbiota

In the line 152 authors should report median (first Quartile- third Quartile) instead of M (P25, P75)

In the table 1, the gestational age of the newborns should be declared in weeks and not in days

In the table 1, authors should be use the term mothers and not mother

In the abstract authors should be not report acronyms. All terms should be reported completely. All the acronyms should be reported only in the text.

In the line 180, authors should also report the composition of unconjugated and conjugated bile acid in the UCB


In the line 200, authors reported that the content derived from CA was substantially lower than from CDCA regarding SBA in meconium. Authors should perform the same analysis for UCB.

In the lines 258-260, authors declare that: the results of this study indicate that, in healthy infants, the composition of bile acid metabolites in meconium differs significantly from that in UCB, with SBAs predominating in meconium. However, authors did not perform a statistical analysis to demonstrate this statement. In the lines 161-179, authors reported the composition of PBA and SBA both in UCB and meconium however did not perform a statistical analysis with p_value.

In the lines 296-297, authors declared that: However, a previous report has indicated that bile acids in UCB can be influenced by bile acids in maternal circulation. I suggest inserting this statement in a more greater context describing the correlation between mother and newborn’s metabolomic profile since pregnancy (Perrone S, Laschi E, De Bernardo G, et al. Newborn metabolomic profile mirrors that of mother in pregnancy. Med Hypotheses. 2020;137:109543. doi:10.1016/j.mehy.2019.109543).

In the line 297 there is a typo error. Please report: It is instead of It’s.

Validity of the findings

The manuscript is original and well conducted.

Authors declared in the paragraph: Statistical Analysis Methodology that p_value threshold was 0,05. Therefore, only p-value<0,05 should be reported. All others p_value thresholds reported are not valid (for example p<0,01 should be replaced with p<0,05)

·

Basic reporting

The article must be written in English and is conform professional standards.

The autors should include articles addressing primary and secondary bile acids, taking into consideration the influence of the maternal metabolome on the newborn.

The structure of the article is conform to an accetable format of standard sections.

The manuscript self contined results relevent to the Hypotesis

Experimental design

In the lines 87-89, authors declare that “Establishing the bile acid metabolomic profile of healthy newborns serves as a foundation for future research in this field. The fetal period, as a unique phase in life, is dynamically influenced by liver metabolism [18] and the mother’s bile acids [19] in terms of bile acid metabolomics.” I am agree with this statement, furthermore the authors can further reinforce this message by also reporting that there is a correlation between the newborn's metabolome and that of the mother since pregnancy, suggesting that the newborn is plausibly “programmed” by the maternal metabolism (Perrone S, Laschi E, De Bernardo G, et al. Newborn metabolomic profile mirrors that of mother in pregnancy. Med Hypotheses. 2020;137:109543. doi:10.1016/j.mehy.2019.109543).

In the line 113, authors should report also Apgar Score of the newborns as a generic marker of good health

Given that the following is a pilot study, it would be helpful if the authors also reported the effect size for each comparison reported. The calculation of the effect size will help other researchers calculate the sample size.

In the table 1, authors should be use the term newborns and not fetus.

In the line 168-169, authors used the term T-conjugate and G-conjucate however these terms should be reported completely. For example, Taurine-conjugate?

In the Line 164-165, authors stated that concentrations of PBA were grater than SBA. However, this statement should be followed by statistical test analysis and p_value. Did author compared with a statistical test the difference between PBA and SBA in both samples?

In the Line 192-193 authors should report p_value of this comparison

In the lines 197-199, author reported ratio CA to CDCA-derived PBA in UCB and meconium. However, probably authors made the ratio CA/CDCA for UCB and CDCA/CA for meconium. Indeed, authors declared “In meconium, the PBA content derived from CA was notably lower than from CDCA [1065.83 (486.68, 1560.61) nmol/g vs 2022.57 (652.54, 3257.16) nmol/g, P < 0.001], so it is not possible that the ratio CA/CDCA is greater than 1.
The relationship between CA/CDCA must be the same both for UCB and meconium.

In the lines 246-248, authors declared that: However, a strong correlation was observed between PBAs and SBAs in meconium, suggesting that the fetal intestine possesses the capability to synthesize SBAs. I suggest removing the fuzzy term “strong”, because in statistic the adjective should be not used to reinforce a statement.

Validity of the findings

No comment

---

## Round 0.2 · accepted · Accept

All issues pointed by the reviewers were addressed and the revised manuscript is acceptable now.

·

Basic reporting

No comment

Experimental design

No comment

Validity of the findings

No comment

Additional comments

The manuscript is well-structured and provides valuable insights into bile acid metabolism in newborns, filling a gap in literature. The revisions address reviewer comments, and the study offers strong potential for future research.

·

Basic reporting

The manuscript meet the standards of the journal. I have no other comments.

Experimental design

The manuscript meet the standards of the journal. I have no other comments.

Validity of the findings

The manuscript meet the standards of the journal. I have no other comments.

Additional comments

I have no other comments, the manuscript can be publish in the current form.

·

Basic reporting

no comment

Experimental design

no comment

Validity of the findings

no comment